# Bladder Cancer during Pregnancy: A Review of the Literature

**DOI:** 10.3390/jpm13091418

**Published:** 2023-09-21

**Authors:** Angelis Peteinaris, Paraskevas Perros, Ioannis Prokopakis, Zacharias Fasoulakis, Thomas Ntounis, Konstantinos Daglas, Ira Eirini Kostopoulou, Athina A. Samara, Konstantinos Pagonis, Vasileios Tatanis, Gabriel Faria-Costa, Rudi Xhaferi, Karen Arzumanyan, Begoña Ballesta Martínez, Athanasios Chionis, Vasilios Pergialiotis, George Daskalakis, Emmanuel N. Kontomanolis, Antonios Koutras

**Affiliations:** 1Department of Urology, University of Patras Medical School, 26504 Patras, Greece; peteinarisaggelis@gmail.com (A.P.); pagonisk7@gmail.com (K.P.); tatanisbas@gmail.com (V.T.); 21st Department of Obstetrics and Gynecology, National and Kapodistrian University of Athens, General Hospital of Athens ‘ALEXANDRA’, Lourou and Vasilissis Sofias Ave., 11528 Athens, Greece; paris_per@yahoo.gr (P.P.); ioannisprokopakis@gmail.com (I.P.); hzaxos@gmail.com (Z.F.); thomasntounis@gmail.com (T.N.); pergialiotis@hotmail.com (V.P.); gdaskalakis@yahoo.com (G.D.); antoniskoy@yahoo.gr (A.K.); 3Department of Obstetrics and Gynecology, Laiko General Hospital of Athens, Agiou Thoma 17, 11527 Athens, Greece; daglask@gmail.com (K.D.); ath.chionis@yahoo.gr (A.C.); 4Department of Neurology, University of Patras Medical School, 26504 Patras, Greece; costopira@gmail.com; 5Department of Embryology, University Hospital of Larissa, Mezourlo, 41110 Larissa, Greece; 61st Department of Urology, Local Health Unit of Matosinhos, 4464-513 Matosinhos, Portugal; gabrielfc60@gmail.com; 7Unit of Oncologic Minimally-Invasive Urology and Andrology, Department of Urology, Careggi Hospital, University of Florence, 50134 Florence, Italy; rudi.xhaferi@unify.it; 8Department of Urology, Yerevan State Medical University Named after M. Heratsi, Yerevan 0025, Armenia; dr.karen.arzumanyan@gmail.com; 9University Hospital del Vinalopo, 03293 Alicante, Spain; bballestamartinez@gmail.com; 10Department of Obstetrics and Gynecology, Democritus University of Thrace, Vasilissis Sofias Str. 12, 67100 Alexandroupolis, Greece; mek-2@otenet.gr

**Keywords:** bladder cancer, pregnancy, cancer treatment, maternal–fetal health, prognosis

## Abstract

Bladder malignancy represents the fourth most common cancer in men and the eighth in women in the western world. Women under 75 years of age have a risk of 0.5–1% of developing bladder cancer. The diagnosis usually occurs between 65 and 70 years of age, whereas the mortality rate for women varies from 0.5 to 4 per 100,000 every year. Nulliparous women present a greater risk than women who have given birth. The risk is further decreased when parity increases. Theoretically, hormonal changes occurring during pregnancy play a protective role. Smoking and occupational exposure to specific chemicals are the most common risk factors of bladder cancer. Other risk factors such as chronic urinary tract inflammation, cyclophosphamide, radiotherapy, and familial correlation have been reported. The aim of this review is to highlight a rare combination, which is the co-existence of bladder malignancy and pregnancy. We present thirteen different cases of women who were diagnosed with malignant bladder tumors during their pregnancy. A review of the literature was conducted, focusing on the unspecific symptoms, possible diagnostic tools, and suitable treatment modalities. The management of bladder cancer in pregnancy is a challenging process. The fragile balance between the possible complications of pregnancy and maternal health is yet to be discussed.

## 1. Introduction

Bladder malignancies represent the fourth most common type of cancer in men and the eighth in women in the western world, with approximately 550,000 new cases annually, worldwide, and recurrences are common [1]. Recent data show that women under 75 years of age are at a risk of 0.5 to 1% of developing bladder cancer. The mean age at diagnosis is usually between 65 and 70 years [2]. The mortality rate varies from 0.5 to 4 per 100,000 every year for women [3].

Studies show that men are in an advanced stage at the time of diagnosis. Nevertheless, survival rates are better in comparison to women; however, the frequency of bladder cancer is increased in men (about 3–4 times) [4]. Published data also report that in nulliparous women, the presence of bladder cancer is greater than in women who have given birth, and the risk is further decreased when parity increases.

Bladder cancer has been undeniably associated with smoking and occupational exposure to specific chemicals [5]. Moreover, other risk factors such as chronic inflammation of the urinary tract, cyclophosphamide, and radiotherapy have also been reported, while familial correlation is a rare phenomenon [6,7]. When it comes to pregnant women, the hormonal changes occurring during pregnancy might play a role [8,9].

The unique characteristics of the urothelium, mainly the layer of the umbrella cells, render both the diagnosis and the treatment of bladder cancer challenging, mainly when combined with pregnancy [10].

The purpose of this review is to unveil the published cases of the uncommon and potentially life-threatening combination of pregnancy and bladder cancer. We hereby present thirteen different cases of women who were diagnosed with bladder cancer during their pregnancy.

## 2. Materials and Methods

For the narrative review and mini case series, we conducted a comprehensive search using MEDLINE (National Library of Medicine, Bethesda, Maryland, MD, USA; January 1980 to June 2023) and the Cochrane Register of Controlled Trials (The Cochrane Collaboration, Oxford, UK). Moreover, we conducted an electronic search using the search terms: “bladder cancer” and “pregnancy”.

To find further research of interest, references of the selected publications and review articles were evaluated. We evaluated all the citations returned from the computerized search, incorporating the following exclusion criteria: studies not related to bladder cancer, non-English language studies, and animal studies. The studies were screened depending on the inclusion criteria and relevant data on study characteristics. Finally, the full text of relevant articles was carefully read and analyzed.

Outcomes were extracted using a standardized pro forma. The inclusion criterion in all studies in this review was the diagnosis of bladder malignancy during pregnancy.

## 3. Case Series

We collected and present all the published cases of pregnant women with malignant bladder tumors. It is important to focus on the symptoms that forced women to present at the hospital, the diagnostic algorithm that health workers used, and the diagnostic dilemmas that turned out to be associated with the pregnancy. Finally, we present the available treatment modalities and how these were adapted to pregnancy, being a unique condition. All the case studies, with the survival outcome after diagnosis and their surgical management, are also presented in Table 1. Case presentation starts with low-grade non-invasive urothelial carcinomas (Ta/T1) that require more conservative strategy treatments.

### 3.1. Case 1

An interesting case has been reported by Castrillo et al. The patient was a 27-year-old female in her 18th week of gestation. Macroscopic hematuria was the reason for admission and the diagnosis of the tumor started with ultrasound and cystoscopy. The patient underwent a transurethral tumor resection procedure under epidural anesthesia, and the surgery was successful. The biopsy report revealed papillary urothelial neoplasm of low malignant potential. In addition, no complication related to the continuation of the pregnancy was reported [11].

### 3.2. Case 2

Mitra et al. reported the case of a 37-year-old female. The patient was in the 11th week of her pregnancy and had a history of occasional recurrent hematuria for the last 16 years. The hematuria was occasionally treated with antibiotics, and sometimes it settled on its own. No further medical history was reported. The patient stated that her hematuria was more frequent during pregnancy. Cytology revealed squamous and urothelial cells. The ultrasound demonstrated a tumor at the bladder base, and the cystoscopy led to the diagnosis of a papillary lesion near the right ureteral orifice and a smaller lesion near it. The medical team and the patient decided to postpone the urological surgery after the delivery. The patient had a delivery in the 40th week of her pregnancy. After 6 weeks, she underwent a transurethral bladder tumor resection followed by mitomycin instillation. The histopathology report demonstrated a non-invasive, papillary transitional cell carcinoma. The 4-month cystoscopy follow-up demonstrated a recurrence, which was treated by electrocauterization and mitomycin instillations [12].

### 3.3. Case 3

The case of a 34-year-old woman in her 20th week of pregnancy was reported by Muezzinoglu et al. The patient demonstrated asymptomatic bacteriuria, without hematuria. Three different microorganisms were isolated in her urine cultures at the 20th, 22nd, and 26th weeks of her pregnancy. Ultrasound examination revealed a lesion of the right bladder wall. A cystoscopy confirmed the tumor diagnosis. A transurethral bladder tumor resection was conducted. The histopathology report demonstrated a non-invasive low-grade urothelial carcinoma. The delivery was completed in the patient’s 40th week of gestation. After the 2-year follow-up the patient was free of disease [13].

### 3.4. Case 4

Shrotri et al. reported a case of a female aged 36 years. The patient was in her 26th week of pregnancy. Her medical urological history included a reimplantation of the left ureter because of vesicoureteral reflux. A gross hematuria forced the individual to present to the hospital. A mass of approximately 2.5 cm on the right bladder wall was diagnosed by ultrasound and cystoscopy. The patient underwent a transurethral bladder tumor resection in the 29th week of gestation. The histopathological report demonstrated a non-invasive transitional cell carcinoma. No recurrence was observed in the first 6 months of follow up [14].

### 3.5. Case 5

This case was reported by Tyagi et al. It was about a woman aged 30 years, with free medical history. The patient was in the 19th week of pregnancy with twins. The routine ultrasound check-up revealed a small bladder lesion. The patient refused the cystoscopy examination. In her 23rd week, she was admitted to the hospital because of occasional recurrent gross hematuria and an ultrasound revealed the same lesion. A transurethral bladder resection was conducted, and 2 papilliferous tumors were resected. The histopathological report demonstrated a low-grade transitional cell carcinoma. The patient went to labor at 35.5 weeks. No ultrasonographic recurrence was found after the labor. Follow-up cystoscopy was suggested [15].

### 3.6. Case 6

Spahn et al., reported a case about a 36-year-old pregnant woman in her 34th week. Occasional vaginal bleeding made the patient visit the hospital. A bladder tumor arising from the right bladder wall was detected by ultrasound. The patient underwent a transurethral bladder resection, and the histopathology report demonstrated a low-grade non-invasive transitional cell carcinoma. The infant was delivered in the 39th week of pregnancy. A follow-up cystoscopy was conducted after one month, with no recurrence. There was no intravesical installation given. A cystoscopy follow-up did not reveal a recurrence for the following year [16].

### 3.7. Case 7

Another case by Spahn et al. was about a 35-year-old woman in her 12th week of pregnancy. This patient underwent a cystoscopy in a follow-up examination after a transurethral bladder tumor resection of a transitional cell carcinoma of the bladder. Recurrence of the tumor was diagnosed. Resection of the tumor was conducted, and a multifocal pTaG2 bladder carcinoma was demonstrated by the histopathology report. A second-look surgery at the 17th week of gestation was necessary based on the multifocality of the mass, without recurrence observation. The patient gave birth in the 40th week of pregnancy, and after 8 weeks a follow-up cystoscopy revealed another recurrence. The biopsy was the same. Intravesical installation was refused by the patient [16].

### 3.8. Case 8

Alleemudder et al. presented the case of a pregnant woman in her 2nd trimester of gestation. The individual was 39 years old and had no other reported health issues. The patient was presented with gross hematuria without abdominal pain. The diagnostic workup was initiated with a sonographic investigation, which revealed a mass of 10 cm in size inside the bladder. The patient underwent a cystoscopy under general anesthesia, and the lesion was removed. The histopathological report revealed that this lesion was a squamous cell carcinoma. Further imaging investigations with MRI were conducted for staging. MRI reports revealed a localized extension of the lesion, and the patient underwent a premature delivery (32 weeks). After that, she underwent a radical cystectomy with an ileal pouch, and the biopsy confirmed the local advancement of the cancer, with negative lymph node specimens. The patient died as a local recurrence of the disease was reported 6 weeks after the surgery, and the progression was inevitable [17].

### 3.9. Case 9

A female aged 27 years with squamous cell carcinoma was reported by Church et al. This patient had no health issues except for recurrent and persistent urinary tract infections. During pregnancy, the patient suffered from recurrent UTIs and microscopic hematuria. The patient was admitted to the hospital because of abdominal pain. Some cardiotocographic abnormalities led to an emergency delivery, to exclude the possibility of an abrupted placenta. Due to catheter blockages and a retention episode after its removal, the patient underwent an ultrasound that revealed thickness and irregularity of the anterior bladder wall. The patient was discharged and was given a cystoscopy appointment 6 weeks later. Some days before the outpatient appointment she was admitted to the hospital with sepsis, suffering from acute renal failure and elevated serum calcium levels. She underwent a CT scan, which revealed an 8 × 12 cm calcified mass in the bladder, thought to be a calcified hematoma. In order to investigate this finding, the patient underwent a cystoscopy, which showed the existence of a tumor in the bladder. A radical cystectomy was conducted. The histopathological report was a squamous cell carcinoma with staging T4N2M0. A local recurrence (11 × 8 mass) that was damaging the left pubic bone was observed 4 months after the surgery. Unfortunately, the patient died 7 months after the initial diagnosis of the malignancy [18].

### 3.10. Case 10

Another case was presented by Rojas et al. A 31-year-old woman without a urological medical history presented with urinary tract infections during the first six months of her pregnancy. The patient was admitted to the hospital presenting a gross hematuria. A cystoscopy was conducted that demonstrated a 3–4 cm tumor of the right bladder wall, which did not allow for the identification of the right ureteral orifice. A 6.3 × 3.5 × 6 cm mass near the ureter and lymph nodes in the right external and internal iliac vessels was revealed by an MRI examination. The patient underwent a transurethral bladder tumor resection, and the biopsy demonstrated an invasive bladder squamous cell carcinoma. No metastatic disease was found by the staging workup. After an oncological counsel and a discussion with the patient, it was decided that the surgical treatment of the bladder tumor would be completed after the delivery, which occurred with a c-section at the 30th week of pregnancy. One month later, the patient underwent a radical cystectomy with bilateral lymph node dissection and a Studer neobladder. The individual had no recurrence after 18 months of follow-up [19].

### 3.11. Case 11

Spahn et al. also reported a case about a 35-year-old woman in the 18th week of her pregnancy. The patient presented occasional macroscopic hematuria that was thought to be a urinary tract infection and was treated with antibiotics. An ultrasound demonstrated a 2 × 4 cm mass in the bladder. A transurethral bladder resection was decided. The histopathology report showed a muscle-invasive grade 3 transitional cell carcinoma with squamous cell metaplasia. The management involved the pregnancy being terminated at the 18th week and staging work up was conducted without the reveal of metastatic disease. The staging was decided as T2N1M0. The patient underwent radical cystectomy and a Mainz-Pouch II diversion after 3 weeks, and a histopathology report revealed an undifferentiated multifocal transitional cell carcinoma with metaplastic squamous cell carcinoma with the invasion of the perivesical fatty tissue, and metastasis in 5 out of 23 dissected lymph nodes. The patient died 2 months after the surgery, as pulmonary metastatic disease was diagnosed. The 2nd and the 3rd case were about smoking patients [16].

### 3.12. Case 12

This case involves a 32-year-old pregnant woman, in her 26th week of pregnancy, who experienced severe blood in her urine (hematuria) for about 3 months. The patient underwent diagnostic procedures, including ultrasound and CT, which revealed the presence of a 5 × 5 cm sized bladder tumor and identified any abnormalities. Additionally, biopsies were performed to obtain tissue samples from the bladder for pathological examination, which confirmed the diagnosis of bladder carcinoma. The rarity of a bladder carcinoma during pregnancy posed a significant challenge in deciding the most appropriate treatment strategy. The patient underwent a transurethral resection of the bladder tumor, a minimally invasive surgical procedure, and 5 weeks later, in her 34th week, gave birth to a 2.14 kg female singleton. The woman was free of disease after 1 year of follow-up [20].

### 3.13. Case 13

This case report presented a 25-year-old pregnant woman who was diagnosed with a significant bladder tumor during her last third of her pregnancy. During the 36th week, due to painless massive hematuria, the patient underwent cystoscopy that revealed a 2 × 2 cm bladder mass. She had a transurethral resection of the bladder tumor, due to a life-threatening hemorrhage, and two days later, an emergency caesarian section due to fetal stress. Finally, biopsy revealed transitional cell carcinoma of the urinary bladder of TaG1 grade, while the 3-month follow-up was clear [21].

**Table 1 jpm-13-01418-t001:** Overview of case studies and survival outcome.

Author	Age of Pregnant Woman	Main Symptom	Week of Gestation Diagnosed	Pregnancy Outcome	Survival Outcome
Castrillo et al. [11]	27	Painless Macroscopic Hematuria	18th week	Full term	No complications in pregnancy were reported.
Mitra et al. [12]	37	Occasional Recurrent Hematuria	11th week	Delivery 40 w	Local recurrence after 4 months.
Muezzinoglu et al. [13]	34	Asymptomatic Bacteriuria without Hematuria	20th week	Delivery 40 w	No recurrence was observed at the 2-year follow-up.
Shrotri et al. [14]	36	Gross Hematuria	26th week	Emergency Caesarian section at 37^+2^ w	No recurrence after 6 months of follow-up.
Tyagi et al. [15]	30	Occasional Recurrent Gross Hematuria	19th week	Delivery 35.5 w	No ultrasonographic recurrence was found after the labor.
Spahn et al. [16]	36	Occasional Vaginal Bleeding	34th week	Delivery 39 w	No recurrence after 12 months of follow-up.
Spahn et al. [16]	35	No symptoms	12th week	Delivery 40 w	Recurrence after 2 months.
Alleemudder et al. [17]	39	Painless Macroscopic Hematuria	2nd trimester	Premature delivery (32 w)	Patient died—local recurrence of disease reported 6 weeks after surgery.
Church et al. [18]	27	Recurrent and Persistent Urinary Tract Infections, Microscopic Hematuria, Abdominal Pain	antepartum	Primary emergency delivery at 37 w	Patient died—local recurrence of disease reported 4 months after surgery.
Rojas et al. [19]	31	Urinary Tract Infections—Gross Hematuria	6th month	Caesarian section at 30 w	No recurrence after 18 months of follow-up.
Spahn et al. [16]	35	Occasional Macroscopic Hematuria	18th week	Pregnancy termination at 18 w	Patient died 2 months after the surgery.
Singh et al. [20]	32	Painless Macroscopic Hematuria	26th week	Delivery 34 w	No recurrence after 12 months of follow-up.
Singh et al. [21]	25	Painless Macroscopic Hematuria	36th week	Emergency caesarian section at 36^+2^ w	No recurrence after 3 months of follow-up.

## 4. Discussion

Pregnancy is a complex condition, with many concerns and challenges both for the obstetrician and the couple. However, bladder cancer during pregnancy consists of a specific and unusual scenario for every specialist, thus requiring the cooperation of urologists, oncologists, psychologists, and other healthcare professionals [22,23].

This special and rare combination sets new challenges, both for the diagnostic and the treatment options, as the state of pregnancy sets new standards and limitations. For example, radiation exposure and the use of anesthesia have to be implemented with careful consideration [24].

The first challenge is the evaluation of common symptoms. As we see in the presentation of the cases, the main symptom was hematuria in almost all of them [25]. Hematuria can be easily misinterpreted as a normal consequence of the pregnancy’s physiological changes and its predisposition of developing urological complications. So, it is considered important for clinicians to be highly suspicious of a more serious underlying condition, especially when hematuria is remarked and symptoms persist with no coexistence of pain. The other common symptom presented is the recurrent bladder infections (case: 3, 9, 10, 12), a state that is also frequent during pregnancy and can disorientate the clinician from the correct diagnosis [26].

Meanwhile, difficulties are increased when it comes to the diagnostic exams. As it is known, the gold standard examination for the diagnosis is cystoscopy, which is an invasive technique that is accompanied by numerous side effects [26]. This limitation gave rise to the necessity for alternative diagnostic tools. Ultrasound can be a valuable tool, however it is not always diagnostic, while the option of CT scanning presents a major disadvantage, that is, radiation exposure for pregnant individuals [27,28].

These limitations make the clinicians more conservative, as the only tool with no contraindication is the use of ultrasound, which comes with limited prognostic value as far as the diagnosis of bladder malignancy is concerned. MRI might be an alternative option, as it can also be safe during pregnancy, however the prognostic value for bladder pathology is limited. The bladder wall consists of four layers: urothelium, vascular lamina propria, muscularis propria, and the outermost serosa. MRI is unable to differentiate the separate bladder layers, urothelium, submucosal layers, detrusor, and perivesical fat reliably. Moreover, both these imaging diagnostic tools do not offer the ability to perform a biopsy, compared to cystoscopy [27].

It is important for the clinicians to be capable of balancing the optimal treatment for the mother, while ensuring the well-being of the developing fetus. Radical surgeries or systematic therapies are not possible during pregnancy, and the balance between disease progression and the maintenance of the fetus’ health is fragile. In all the cases above, it is obvious that when the diagnosis was set up primarily, not only the prognosis was far better, but the patient also experienced a less radical treatment with less morbidity, compared to a delayed diagnosis and a more radical therapy with advanced outcomes, both for the mother and the fetus (case: 8, 9, 10, 12, 13) [29]. However, the selection of treatment modalities should be individualized, considering factors such as tumor characteristics, gestational age, and maternal health.

Another remarkable aspect is the psychological burden. The fragile psychology of a pregnant woman is heavily affected by the unexpected news of cancer and its management. The emotional impact of a cancer diagnosis, the stress of making treatment decisions, and concerns about fetal well-being can significantly affect the pregnant woman and her family. Throughout the entire process, co-operation with specialists is required, so as to support and counsel the patient and her relatives.

Notably, the available data on the epidemiology of bladder cancer during pregnancy are limited, primarily due to its rarity. Prospective studies are necessary in order to accurately evaluate the incidence and the prevalence of this rare combination. Furthermore, the most important field of research must be the identification of potential risk factors, clinical outcomes, and the comprehension of the underlying mechanisms leading to the development of bladder cancer during pregnancy that might contribute to early detection and targeted interventions. As a consequence, it is important for clinicians to carefully obtain every patient’s medical history, bearing in mind the prognostic factors of bladder malignancy.

For example, about 50% of bladder cancer cases are tobacco-related, while smoking raises bladder cancer risk by two to three times. Moreover, the percentage of bladder cancer risk associated with previous smoking is up to 44%. Bladder cancer risk is positively proportional to the smoking intensity of up to 20 cigarettes per day but plateaus beyond that. However, smoking duration raises bladder cancer risk without plateauing. Bladder cancer risk lowers after quitting smoking. Additionally, it is found that lifetime second-hand smoking exposure also increases bladder cancer risk in non-smoking individuals. [5].

As a result, a female smoker who might be under chemotherapy or radiotherapy presenting with urinary bleeding has to be evaluated with high suspicion. Moreover, prenatal counseling should always include advice about ceasing smoking and avoiding exposure to chemicals. As for urinary tract infections, they are suggested to be treated too, and if recurrences occur accompanied by other clinical manifestations of bladder malignancy, suspicion should be raised. Lastly, clinicians ought to evaluate the potential risks and benefits when the need to use cyclophosphamide and radiotherapy occurs (Figure 1).

## 5. Conclusions

The overall incidence of malignancies during pregnancy is rather uncommon and is calculated to be around 2.35/10,000. The synchronous diagnosis of a malignant bladder tumor during a woman’s pregnancy is even rarer and poses a challenging condition that requires the cooperation of many specialists and a multidisciplinary approach for effective management. High suspicion, early diagnosis, appropriate treatment selection, and consideration of the unique circumstances of each case are crucial for optimal outcomes. The patients should take part in the decision of the management and the different therapeutic options, after being informed about the possible scenarios for their health. Due to the condition’s rarity, the literature data are limited, thus continued research and collaboration among healthcare professionals are essential to improve outcomes for both the mother and the developing fetus, improving diagnostic and treatment strategies.

## Figures and Tables

**Figure 1 jpm-13-01418-f001:**
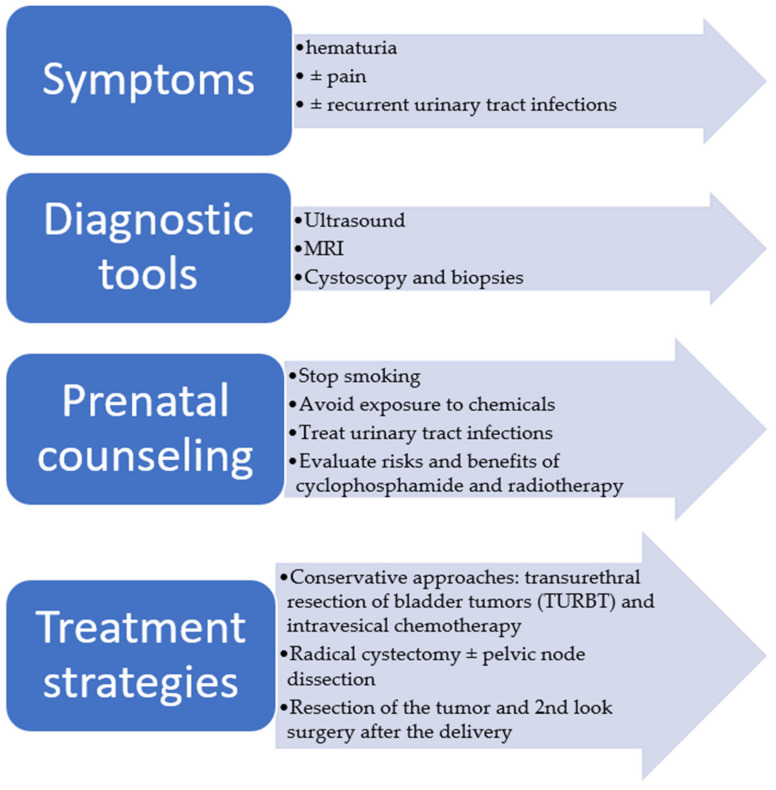
Overview of symptoms, diagnostic tools, counseling, and treatment options.

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
