# Peer review of "Bladder Cancer during Pregnancy: A Review of the Literature"

_jpm, 2023, doi:10.3390/jpm13091418_

Round 1

Reviewer 1 Report

Thank you for submitting your work to JPM. I read the paper with interest, and I think it is generally a well-written text, that addresses an important (though infrequent) clinical situation. However, I do have some comments and suggestions for further improving, should you consider taking them into consideration:

Inclusion/exclusion criteria for the cases presented are somewhat unclear. The most important observation is that the authors should decide whether they wanted to include all bladder tumors in their research, or just malignant ones. The title states "bladder tumors", and this term is used on several occasions in the text - suggesting ALL tumors, including benign, were considered; however, the search terms themselves were "bladder cancer", and "studies not related to bladder cancer" was listed as an exclusion criterion. If malignancy was indeed the research target, then some reported cases diagnosed with benign, borderline or undefined lesions should be removed (e.g., cases #1, #5). Also, some patients were not newly diagnosed during pregnancy, but rather had a relapse of a previous malignancy (e.g., case #6) and should perhaps be reported separately, as they do not raise the same diagnostic difficulties. As a suggestion, I would have also grouped low-grade non-invasive urothelial carcinomas (Ta/T1) separately, as their treatment involves TUR-B alone for most cases (even outside pregnancy); it is generally agreed that bladder instillations, if indicated, could and should be postponed for after the pregnancy is carried out. Also, even if published by the same authors, each case should be reported separately also in the text (not only in the summary table), to avoid reader confusion (e.g., case #11). I known, information on pregnancy outcome (long-term if possible) should also be included in the text/table.

Discussions should be extended to include comments on some more points of particular interest (e.g., patient #6, who was administered BCG right before pregnancy; or the cases with squamous histology). The last paragraph of the section should be merged with the Conclusions.

English use is adequate, however the authors might consider rephrasing a few sentences (e.g., pg.2/ln.61: "[...] to amass the published cases of the rare and life-threatening combination of women who are pregnant and sometime during the pregnancy, suffer from bladder cancer"; pg.9/ln.292: "Pregnancy is a physical but complex situation [...]")

Author Response

Dear reviewer,

Thank you for your kind remarks and suggestions.

Firstly, we decided to include only the cases of newly diagnosed bladder cancer, so as a result, we removed the cases of benign, borderline or undefined lesions, as well as some patients that were not newly diagnosed during pregnancy, but rather had a relapse of a previous malignancy.

Additionally, we have also grouped low-grade non-invasive urothelial carcinomas (Ta/T1) separately, and reported each case, even when written by the same author separately to avoid reader confusion. Also, information on pregnancy outcome was included in the table.

Discussion was extended and the last paragraph of the section was merged with the conclusions.

We also edited the text for any language mistakes, for better understanding.

Yours sincerely,

Athina Samara

Reviewer 2 Report

This is a review of a rare disease in pregnancy. The authors set out the challenges with early diagnosis and treatment. The presence of frank, painless hematuria and recurrent urinary tract infections not responding to treatment are warning signs of possible bladder cancer. Despite the risks of cystoscopy and imaging, investigations are crucial for early diagnosis.

Is there a role for MRI? This is safer in pregnancy than CT scan.

For the authors, as a busy practitioner- I would love to see key messages highlighted in a box. The role of smoking and Bladder cancer should be emphasized more in the discussion.

You review will add to knowledge in the management of this rare disease in pregnancy.

The manuscript generally would require editing for language for better understanding.

Author Response

Dear reviewer,
Thank you for your kind remarks and suggestions.
We tried to incorporate your comments, by adding a part concerning the role of MRI, as
suggested.
Additionally, we included the key messages of the review, highlighted in a box, as well as a
section for the role of smoking and bladder cancer in the discussion.
We also edited the text for any language mistakes for better understanding.
Yours sincerely,
Athina Samara